# Animal behavior is central in shaping the realized diel light niche

N. Sören Häfker [1,2 ✉], Stacey Connan-McGinty[3], Laura Hobbs[4,5], David McKee [3], Jonathan H. Cohen [6] & Kim S. Last[5]

Animal behavior in space and time is structured by the perceived day/night cycle. However, this is modified by the animals' own movement within its habitat, creating a realized diel light niche (RDLN). To understand the RDLN, we investigated the light as experienced by zooplankton undergoing synchronized diel vertical migration (DVM) in an Arctic fjord around the spring equinox. We reveal a highly dampened light cycle with diel changes being about two orders of magnitude smaller compared to the surface or a static depth. The RDLN is further characterized by unique wavelength-specific irradiance cycles. We discuss the relevance of RDLNs for animal adaptations and interactions, as well as implications for circadian clock entrainment in the wild and laboratory.

[1] Max Perutz Labs, Vienna BioCenter, University of Vienna, Vienna, Austria. [2] Research Platform 'Rhythms of Life', Vienna BioCenter, University of Vienna, Vienna, Austria. [3] Physics Department, University of Strathclyde, Glasgow, UK. [4] Department of Mathematics and Statistics, University of Strathclyde, Glasgow, UK. [5] Scottish Association for Marine Science, Oban, UK. [6] School of Marine Science and Policy, University of Delaware, Lewes, USA. ✉email: soeren.haefker@univie.ac.at

The 24 h day/night cycle structures animal behavior and interactions in that for any given action (e.g. resting, foraging, migrating, reproduction) there is an optimal time of the day that depends on the environmental conditions[1]. This concept is embedded in 'adaptive resource ecology' that describes how animals modulate foraging activity in time and space as an adjustment to environmental cycles of food availability/distribution and predation risk[2]. The most prominent cyclic environmental cue over the course of the day is the light/dark cycle, which directly drives temperature cycles and indirectly, foraging success and predation risk. Importantly however, animal behaviors in time directly influence their experience of the surrounding environment. For example, diurnal species encounter very different light conditions (and changes thereof) compared to species that are nocturnal or crepuscular. Consequently, behavior-dependent 'environmental experience' should be reflected in adaptations of an animal's physiology, timing mechanisms and adaptive capacities. Here we discuss this feedback between behavior and the environment in the context of the fundamental versus realized niche theory[3].

The 'fundamental', or hypothetical niche, is the one the organism inhabits, where it can take advantage of all biotic and abiotic factors without competition or predation, much like in a perfect laboratory environment with no predation and food *ad libitum*, i.e. all habitats where physical conditions allow for long-term survival. In the wild however, organisms inhabit 'realized' niches[4], with resource limitation, competition and predation, and where their own behavior and physiology help to shape their specific realized niche, i.e. the environmental conditions directly experienced by an animal. Behavioral and physiological responses to the 24 h day/night cycle that have evolved over long evolutionary time-scales[5], represent one major way the realized niche can form. The diel light cycle is the main cue for the synchronization ('entrainment') of circadian clocks that orchestrate rhythms of behavior and physiology[6]. However, while there are cases of animals switching their diel activity window in response to changes of biotic and abiotic conditions[7–9], the underlying clock rhythmicity tends to be highly conserved and the potential for fast evolutionary adaptation limited[10]. This means that after a switch in activity patterns, conserved clock mechanisms have to keep functioning in a strongly altered realized niche where a transfer of animals to the laboratory (i.e. to a fundamental niche) can result in a rapid switch to the evolutionary conserved activity pattern shaped by the circadian clock[11]. This highlights the complex relationships between fundamental and realized niches, behavioral and physiological rhythms, and underlying endogenous timing mechanisms. With diel light changes being central to animal rhythms of behavior and physiology, as well as the entrainment of circadian clocks, we need to better understand light from an organism's perspective.

Here we use a marine zooplankton community as a model system to investigate diel behavioral rhythms shape the animals realized niche with a special focus on the *Realized Diel Light Niche* (RDLN) and how it differs from light in the 'fundamental' habitat. We described the ecological consequences and benefits of the realized environment and discuss its relevance for the functioning of endogenous clocks and for laboratory investigations.

## Results and discussion
### Marine systems as natural light laboratories.
Marine environments are highly suited for light exposure experiments given the spectral light attenuation with depth and its change from coastal to oceanic waters[12]. In the open water, myriads of small drifting primary and secondary consumers (zooplankton), as well as higher level predators such as fish, move in a 3-dimensional habitat with a vertical and temporal light structure. Reliance on light for predation and its avoidance is fundamental in such ecosystems and drives diel vertical migration (DVM) of zooplankton, which ascend to surface waters at night to feed, and descend back to depth during the day to avoid predators. This visual arms race is exceptionally well documented[13] and is potentially the largest daily migration of biomass on earth[14]. Furthermore, DVM is central to population and food chain structure[15] and biogeochemical cycles[16,17].

Light is considered the dominant proximate cue for DVM[14,18] with animals commonly staying below a changing depth of threshold light intensity termed the isolume[18]. Depth adjustment is often closely connected to light intensity with organisms rapidly responding to solar and lunar light sources[19,20], as well as short-term irradiance changes from cloud cover[21], phenomenological[22] and anthropogenic events[23]. Furthermore, the migration behavior is often under some degree of circadian clock control imparting an innate cue to migrate when environmental signals may be damped, masked, or absent[18,24]. This may create a paradox as the 'isolume-following' migration behavior should mask, or potentially even remove, the overt diel light cycle (in effect shaping the RDLN) which is considered the principle DVM trigger. Indeed, we can now question if the realized light niche of migrating zooplankton is even sufficient to entrain a circadian clock? Or are other parameters that change with depth (i.e. spectral light composition, pressure, temperature, oxygen, etc.) potentially more relevant in the homogenous light environment of the realized niche[25–27]? Marine communities that perform synchronized vertical migrations thus present excellent model systems to study how behavior and environmental conditions affect each other, and how this may shape animal rhythmicity, predator–prey interactions and circadian clock entrainment in mobile animals.

### The realized diel light niche of Arctic zooplankton.
In order to determine the RDLN of Arctic zooplankton performing DVM, we focused on Kongsfjorden in the Svalbard archipelago during the spring equinox, a time when DVM was extant and potential co-factors affecting the light environment (freshwater runoff, phytoplankton concentration) were minimal (Supplementary Fig. 1). The reasons why DVM was present in spite of low levels of phytoplankton are discussed in detail elsewhere[28], but could ensure optimal resource usage once the spring algae bloom starts. We determined the vertical distribution of animals via echosounders (ADPCs), using the zooplankton Center of Mass (COM) as a proxy[29]. Light was measured at the surface and changes in intensity and spectrum with depth were modeled. The fjord has a well-characterized zooplankton community[20,30–32] and during the study period, overt synchronized DVM spanning almost the entire 230 m water column was evident (Fig. 1a). Diel changes in hyperspectral irradiance were corrected for the spectral sensitivity of the dominant zooplankton species (utilized photons, see the "Methods" section) and spanned ~6 orders of magnitude at the surface and at 120 m depth (equating to the middle of DVM range) (Fig. 1b). The RDLN was determined by comparing the diel changes of the modeled underwater light field to the distribution of the migrating zooplankton. At the COM, the RDLN spanned ~4 orders of magnitude (Fig. 1b, c). Comparing the utilized photon intensity ranges, it is evident that changes in RDLN at the COM are 2 orders of magnitude smaller than light changes at the surface or at a constant depth of 120 m (Fig. 1c). Thus, the diel light cycle, which would be experienced by the migrating animals, is strongly dampened, highlighting the large effect of DVM behavior on the realized light environment. Furthermore, zooplankton would have been able to migrate

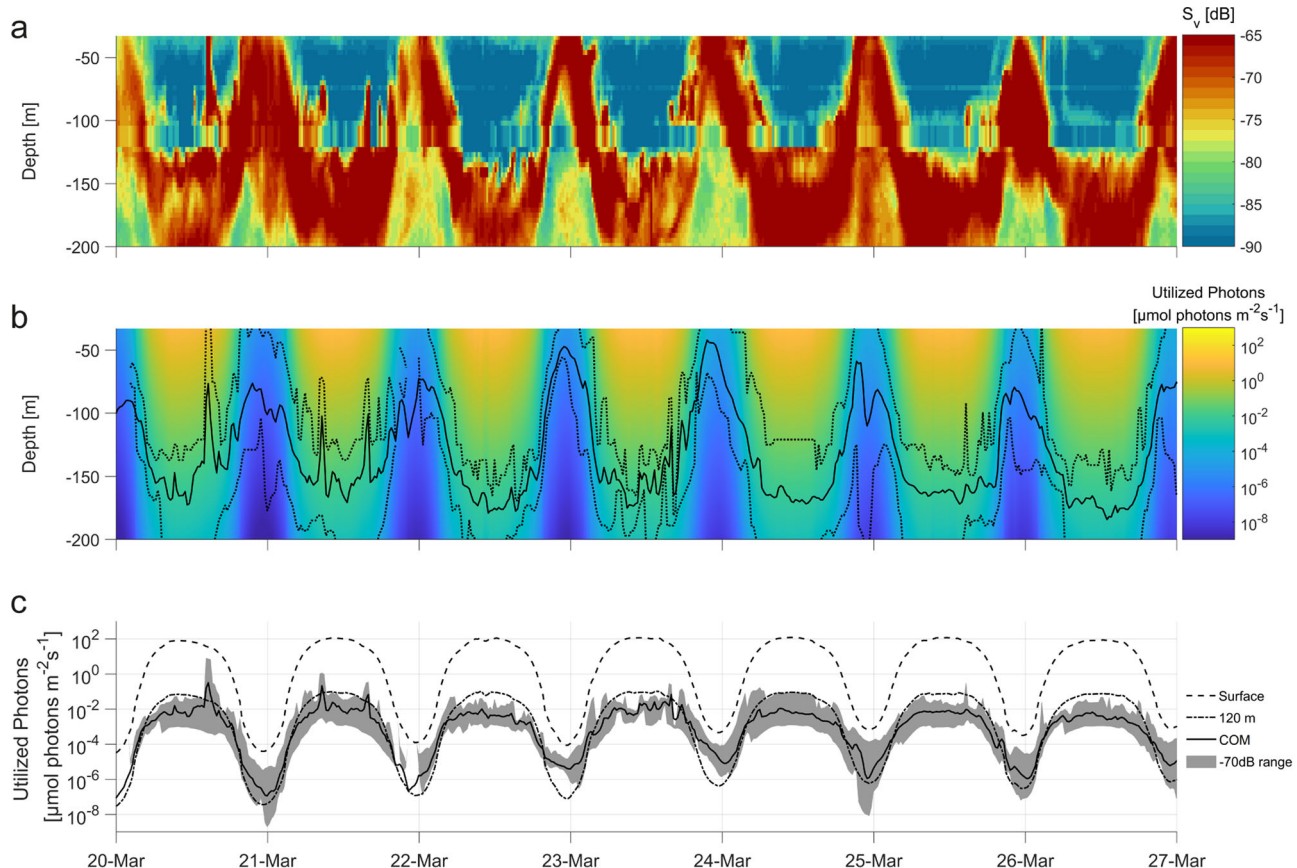

**Fig. 1 Zooplankton DVM and light intensities in Kongsfjorden around the spring equinox. a** Vertical zooplankton distribution as backscatter intensity ($S_v$). The disturbance at ~120 m results from the ADCPs being located at that depth and not being able to measure in direct proximity. **b** Diel light cycle of PAR irradiance and attenuation with depth. Solid black line: zooplankton center of mass (COM), dotted lines: −70 dB backscatter thresholds indicating range of zooplankton aggregation. **c** Logarithmic PAR irradiance just below the surface, at a constant depth of 120 m, and the realized diel light niche at the COM. Gray shading indicates PAR in the −70 dB range shown in panel (**b**). For times where the −70 dB range could not be properly determined, it was left out. Dates on the x-axis indicate the start (midnight) of the respective days.

deeper during the day, meaning that RDLN experienced is directly modulated by behavioral choice.

The influence of migration behavior on the realized light field is also evident from the spectral composition at the COM, which shows dampened rhythmicity across the sensitivity-corrected spectrum (Fig. 2) along with characteristic depth differences due to the wavelength-specific attenuation of light in water (Supplementary Fig. 2). Significantly, the effect of a daytime residence in deeper, blue-dominated water because of DVM, causes exposure of zooplankton to an inversion of the realized diel intensity cycle at wavelengths >570 nm (i.e. higher intensities at night). In contrast, intensity changes at the surface are similar across the entire spectrum (Fig. 2a). Periodogram analysis shows that rhythmic diel oscillations in the light signal at the COM are strongest at 445–535 nm (the wavelengths penetrating deepest in water), and that rhythmicity briefly re-occurs at 590–600 nm with an inverted diel intensity cycle (Fig. 2b, e). It is not clear if zooplankton have sufficient visual perceptions capabilities for the light changes at >550 nm[33], but light may also be perceived non-visually with extra-ocular photoreceptors and at very low intensities[34,35].

Thus, the environment perceived by the migrating zooplankton shows dampened light intensity changes compared to the surface or a static depth, whereas the realized changes in spectral light composition are more complex. While not investigated in detail here, DVM further enhances or suppresses diel cycles in other environmental parameters such as temperature, salinity and

pressure, which may also contribute to rhythmic behavior and provide circadian entrainment cues[25,26] (Supplementary Fig. 3).

**The RDLN in an ecological context.** Although dampened, cycling of the RDLN still occurs during DVM (Fig. 1b, c). However, diel intensity changes at 550–600 nm are minimal (Fig. 2b, d, e). Thus, animal depth selection could be mediated through light receptors sensitive to this spectral range. Additionally, relative spectral intensity changes with depth may provide a depth gauge supporting optimal positioning[36]. Considering the vertical spread of the observed scattering layer (Fig. 1a), it is likely that light intensity sets an upper limit rather than a specific 'target depth' (as seen in Hobbs et al. 2021)[37], which would reduce energy costs of constant depth adjustments. Such an upper limit could reflect the detection threshold of specific zooplankton light receptors or the light intensities at which visual predators can efficiently hunt zooplankton[37].

It must also be considered that the investigated scattering layer consists of a multitude of animals that all occupy different depths and show species- and life stage-specific migration behaviors[30,38]. There are, for example species that, depending on the ecological context, ascend to the surface during the day (reverse DVM)[18,39,40] or move to intermediate depths in the middle of the night (midnight sinking)[18,41], thereby creating different RDLN patterns in multiple scattering layers. As larger individuals are more visible and thus more vulnerable to predation near the

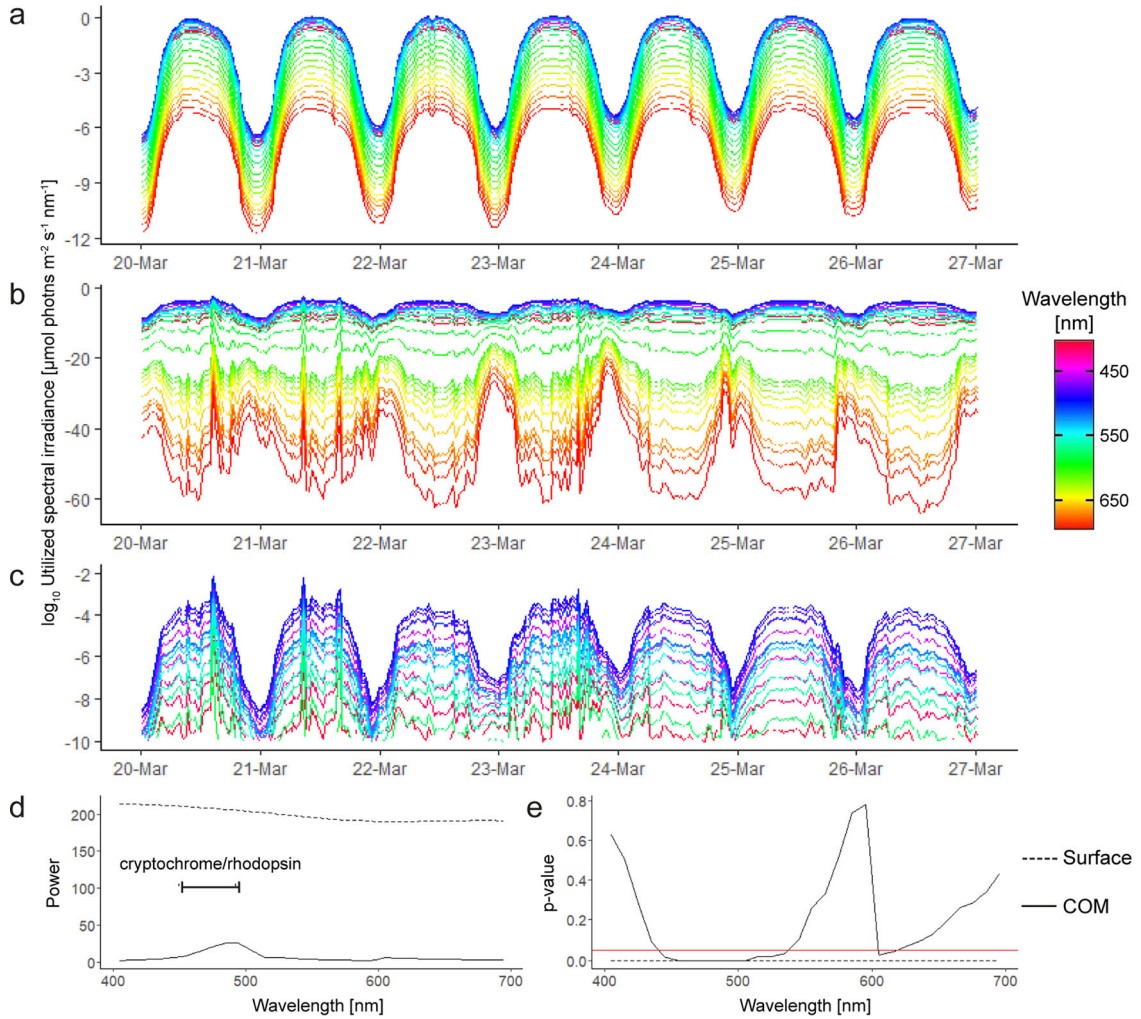

**Fig. 2 Realized spectral light changes. a** Sensitivity-corrected spectral light intensities just below the surface. **b** Sensitivity-corrected spectral light intensities realized at the depth of the zooplankton center of mass (COM). **c** Blow-up of panel (**b**) focusing on the 400–590 nm range. Dates on the *x*-axis indicate the start (midnight) of the respective days. **d** Power and **e** significance of spectral light signal rhythmicity just below the surface and at the COM. Determined through Lomb–Scargle rhythm analysis (see the "Methods" section). The red horizontal line indicates the significance threshold of 0.05.

surface, body size can be a major factor determining migration behaviors and hence specific RDLNs[42,43]. Species further migrate at different speeds and individuals often show periods of intermitted resting or burst swimming[44], meaning that RDLNs are more complex than indicated by simply the COM depth. In general, RDLNs should be investigated on the level of the smallest biological unit possible, ideally in individuals.

Additional benefits of RDLNs could arise from the narrowing or the realized light intensity range as observed here in zooplankton. This should reduce the costs of cyclic light sensitivity adjustments due to the production/degradation of pigments, enabling organisms to optimize their light sensing systems for a specific intensity range. The tendency for an optimization of light sensing systems is illustrated by the variety of light receptors found in different aquatic clades[45,46]. Adaptation to RDLN-related light changes could further help explain circadian clock-controlled rhythms of visual light sensitivity in several species[47–49]. Rhythmic regulation of light sensitivity by a clock does prevent costly responses to short light condition changes that can arise from behaviors such as rapid water depth changes, escape responses (fleeing/hiding), or the leaving/entering of burrows[50–52]. Clock-controlled light sensitivity cycles could further help to prevent responses to the RDLN 'noise' that results e.g. from non-continuous swimming during DVM[44].

The DVM behavior resulting from the trade-off between food availability and predation risk thus creates an RDLN to which the animals have adapted through phenotypic plasticity[40,53–55] as well as evolutionarily[47–49,56]. A detailed description of behavior and RDLNs in the wild will therefore lead to a better understanding of animal fitness, adaptability and ecosystem functioning.

**Circadian clock entrainment by the RDLN**. Realized light rhythmicity is strongest at 445–535 nm (Fig. 2c–e), i.e. the wavelength range to which the dominant zooplankters in Kongsfjorden, Arctic krill (*Thysanoessa* spp.) and copepods (*Calanus* spp.), show the highest photosensitivity[33,57]. Furthermore, cryptochromes, the central pigments for circadian clock entrainment in most invertebrates, are maximally sensitive at this range[58,59]. Circadian clock rhythmicity in zooplankton has been observed in the laboratory[24,60–62] and in the wild[63–65], however, the dominant circadian entrainment cues for zooplankton are not yet described in detail. Light sensitivity studies suggest that the strongly dampened realized light cycles in our study can still be perceived and therefore act as an entrainment cue[19,33,66,67]. Low-light entrainment is also achieved in terrestrial organisms; for example, *Drosophila*'s Cryptochrome 1 can integrate very weak

blue light cues ($3\,\mathrm{nW\,cm^{-2}} = 1.17*10^{-4}\,\mu\mathrm{mol\ photons*m^{-2}*s^{-1}}$) over at least $6\,\mathrm{h}$[35].

Given the active behavioral dampening of the light intensity cycle, it is worth exploring other potential entrainment cues. The RDLN of migrating zooplankton shows distinct diel spectral shifts, highlighting the potential for spectral entrainment of diel rhythmicity (Fig. 2). Spectral changes can be potent entrainment cues in arthropods[68], fish[69] and mammals[70,71] and are considered particularly important at high latitudes and when intensities are variable[70,72,73]. Our data show that RDLN intensity changes are strongly dampened by DVM and spectral light sensing could support entrainment under these conditions. However, whilst DVM and the resulting RDLN may entrain the circadian clock, clock-driven behaviors also affect the RDLN, causing a circular argument[24,74]. This can be resolved as migrations are not exclusively clock-controlled, but also directly respond to light (masking-effect). Behavioral responses can thus evoke realized cycling in light and other parameters, which can then act as entrainment cues. In the DVM-context, the circadian clock may mostly support the anticipation and preparation of vertical migrations (e.g. by adjusting metabolism[24,74,75]). The circadian clock can further ensure proper timing of behavior and physiology when entrainment cues from the RDLN become too weak or variable[76,77].

**From the wild to the laboratory.** A detailed description of RDLNs in the wild can prove beneficial for designing laboratory studies on both terrestrial and aquatic organisms. Field investigations can reveal unexpected environmental features and their effect on behavior[34], and can help to determine realistic ranges of abiotic factors to then investigate in the laboratory. The potential of RDLNs in chronobiological studies is underlined by field work on subterranean rodents, where burrow emergence behavior strongly modifies realized light cycles and indicates complex circadian entrainment processes[52,78,79]. Behavioral rhythmicity differences between laboratory studies and wild or semi-natural conditions are common, which is typically attributed to the artificial conditions in the laboratory[71,80–86]. Where wild investigations are not possible, understanding of RDLNs can help develop experimental procedures that more closely resemble nature and focus on the cues most relevant to the organism in the wild[87]. For zooplankton collected in Kongsfjorden and brought to the laboratory, this would mean implementing laboratory light systems with strongly dampened and gradual intensity changes as well as spectral conditions that mimic the RDLN. Unless sensitivity studies show them to be non-detectable, the light system should also include wavelengths >550 nm and the inversion of the intensity cycle (Fig. 2b). By comparing laboratory RDLN-simulations to field observations and studies under standard laboratory conditions, the environmental cues most important for behavioral responses and circadian clock entrainment can be determined and the sensing/processing pathways can be explored. Simulating RDLNs can further help to determine, if differing behavioral rhythm in the laboratory and the wild are the result of artificial housing conditions, or if they represent the return to an evolutionarily conserved rhythmicity that was masked under the RDLN conditions in the wild, but re-emerges under the fundamental niche conditions in the laboratory[10,11]. Finally, experiments may further be supplemented with heterogeneous housing conditions (e.g. darker/colder hiding compartments)[81,88,89], that would allow for the emergence of natural animal behaviors and a partially re-creation of RDLNs in the laboratory driven directly by the behavior of the study organism.

## Conclusions
We show that the realized diel light niche (RDLN) experienced by vertically migrating zooplankton strongly deviates from classical environmental measurements, resulting in unexpected cyclic patterns of light conditions. While realized diel changes in light intensity are strongly dampened though likely still detectable, the relative intensities of different wavelength show marked diel cycling and may support entrainment of the circadian clock, central to various biological processes. Describing the RDLN allows us to look at the environment "from the animal's perspective" and reveals how profoundly animal behavior can affect the environment it experiences. This is crucial to determine the environmental conditions, which are the driving forces behind an organism's responses, adaptations, and endogenous timing mechanisms. The importance of the natural environment with all of its complexity is increasingly gaining recognition also in the molecular field[90]. Our results highlight the benefits of a detailed description of natural habitats and animal movement therein, both for the investigation of behavior and species interactions in the wild as well as for the design of laboratory experiments.

## Methods
**Study site and zooplankton community**. Data were collected from a mooring in Kongsfjorden (~230 m bottom depth), a fjord in the Arctic Svalbard archipelago (78°57.54'N, 11°49.44'E, Supplementary Fig. 1a). Two 300 kHz Acoustic Doppler Current Profilers (ADCPs) were installed on the mooring at ~120 m, one in an upward and one in a downward configuration. ADCP data were processed to mean volume backscattering strength ($S_v$ [dB])[91]. The biomass of the zooplankton community is strongly dominated by euphausids (krill) of the genus *Thysanoessa* with additional contribution by copepods of the genus *Calanus* (*C. finmarchicus* and *C. glacialis*)[92]. Both groups are well known to perform DVM in the fjord[30,92]. A validation via depth-resolved net-sampling at the time of the study was not possible, but would benefit future studies to gain further insights into species- and life stage-specific RDLNs. To determine the depth range of highest zooplankton biomass throughout the water column, the Center Of Mass (COM) of the backscatter data (as described in Urmy et al. [29]) and a −70 dB threshold (similar to methods applied by Cottier et al. [93]) were used. COM depth was used to extract light, temperature, and pressure data for "realized" conditions. Temperature data were collected using 12 sensors attached to the mooring and distributed through the water column, using a mixture of Seabird 56+, Seabird 37, and temperature mini-loggers. The pressure experienced by the zooplankton was calculated directly from the COM depth (pressure [atm] = 0.0992*depth [m]+1).

We focused our study on the week around the spring equinox in 2018 (20–26 March 2018) with a photoperiod (sunrise to sunset) lengthening from 12 to 14 h during the study period (www.timeanddate.com). The phytoplankton concentration at the time was at its annual minimum and hence it was not included for underwater light calculations (Supplementary Fig. 1b). Rhythmicity of light (see below), temperature and pressure over the study period was determined via Lomb–Scargle periodogram analysis (TSA Cosinor v6.3). The method uses a least-square fit to a sinusoidal curve to describe rhythmicity in the data.

**Light measurement and depth attenuation**. Spectral irradiance above the surface was measured by an USSIMO sensor (In Situ Marine Optics, Australia) at the ArcLight Observatory[94] outside Ny-Ålesund, Svalbard, ~1 km from the mooring site (Supplementary Fig. 1a). Spectral irradiance was recorded at 380–950 nm with 10 nm resolution in 20 min intervals. Measurements in the PAR range (photosynthetic active radiation, 400–700 nm) were extracted for use. Once the sun was >7° below the horizon, irradiance levels failed to exceed the baseline sensitivity of the USSIMO detector (~$10^{-3}\,\mu\mathrm{mol\ photons\ m^{-2}\,s^{-1}}$). Irradiance levels for angles >7° below the horizon were approximated via linear regression on a subset of irradiance measurements recorded between 90° and 97° for each wavelength, selected for cloud and moon free conditions to best represent the ambient light field.

Above surface irradiance was divided into diffuse and direct components and attenuated down the water column according to the method described by Pan and Zimmerman[95] modulated by local cloud cover data obtained from the European Centre for Medium-Range Weather Forecasts (ECMWF) Re-Analysis 5[96]. Direct in situ measurements were not possible due to the currently still limited sensitivity of hyperspectral sensors[97]. The approach has previously been validated using in situ measurements[95] and was further validated by us against underwater PAR irradiance calculated by the Hydrolight radiative transfer software[98]. The light field was transmitted over the air–ocean boundary using Fresnel coefficients for the direct component and a regression analysis function for the diffuse component derived from extensive simulations using Hydrolight[98]. Spectral irradiance was modeled down to 200 m depth at 1 m resolution with a region-specific bio-optical model[99]. To take animal spectral light sensitivity into consideration, overall light intensities were converted to utilized photons using the spectral light sensitivity of the dominant zooplankton species, the krill *Thysanoessa*[33]. For future studies,

validation of modeling using light measurements made below the surface would further improve results[97,100].

**Reporting summary**. Further information on research design is available in the Nature Research Reporting Summary linked to this article.

## Data availability

Raw data associated with the figures is provided online (Supplementary Data 1). Any additional information can be obtained directly from the authors.

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

## Acknowledgements

Support for this work came from the Austrian Science Fund (FWF, http://www.fwf.ac.at/en/) through an SFP grant (SFB F78) and from the Norwegian Research Council project Dee-pImpact (project no. 300333). Additional support came from the CHASE project, part of the Changing Arctic Ocean program, jointly funded by the UKRI Natural Environment Research Council (NERC, project no. NE/R012733/1) and the German Federal Ministry of Education and Research (BMBF, project no. 03F0803A). The ArcLight Observatory is operated by the Arctic ABC Development program funded by the Norwegian Research Council (project no. 245923). N.S.H. was further supported by a Lise-Meitner-fellowship by the Austrian Science Fund (project no. M2820).

## Author contributions

N.S.H., K.S.L., and J.H.C. were responsible for the conceptualization of the study. N.S.H. wrote the manuscript with the support of the co-authors. L.H. handled the zooplankton backscatter data. S.C.-M. and D.M. processed the light data. K.S.L. and J.H.C. performed the rhythm analyses. Graphs were created by L.H., S.C.-M., J.H.C., and N.S.H.

## Competing interests

The authors declare no competing interests.

**Additional information**

