## [Peer Review File · Communications Biology]

Reviewers' comments:

Reviewer #1 (Remarks to the Author):

Summary:

The authors postulate that animals' realized vs. fundamental niche has a strong light component and that the realized light niche is shaped by animal behavior. The authors suggest that to understand the role of light in the realized niche, that the perspective of the organism should be taken. They use a marine zooplankton community to demonstrate that due to diel vertical migration (DVM) that factors experienced as part of the light environment by the zooplankton is rather similar or opposite than expected across the diel cycle depending on depth. The authors conclude that the DVM behavior of the zooplankton affects their realized light niche.

Impression:

I really enjoyed reading this manuscript because the writing was clear and concise. I believe it is a novel contribution to the literature, specifically the concept of taking the viewpoint of the animal's perception of light across the diel cycle. This angle is rarely taken in empirical investigations, especially in the field, and can be useful by many disciplines, including evolutionary biology, ecology, physiology, and chronobiology. This paper specifically highlights the need to incorporate cues important to the organism being investigated to better understand physiological mechanism and ecological process. Altogether, this manuscript would be a great contribution to the literature if relatively few issues were resolved.

Data & methodology, including statistics:

There is no data availability statement, and it is unclear whether the calculation of COM is a result of programming using code or a specific program. The supplied data appear to be good quality and seems efficient to repeat the analyses.

I am unable to evaluate fully the marine zooplankton community and marine irradiance measurements because these methods are outside my expertise. The majority of the methods presented are following previously published methods.

Lomb-Scargle periodogram analysis is an appropriate method for identifying rhythmicity in ecological data.

Comments:

Although, I believe that the realized diel light niche (RDLN) is a useful term in the context of this manuscript, the term draws a strong analogy to previous discussion on the temporal niche, in which animal behavior and species interactions directly influence the light experienced by an organism. The following articles are some seminal discussions on this topic that should be incorporated into the manuscript: Kronfeld-Schor et al. 2001; Kronfeld-Schor and Dayan 2003.

There seems to be some logical inconsistencies in whether the realized niche is a property of an organism resulting from their adaptations (lines 43-45) or is a group of selective forces acting on an organism resulting in adaptations. The definition of "realized niche" in the manuscript states that organisms need to be (i.e., must be) adapted to the realized niche (lines 41-43). Kearney (2006) defines the realized niche as a property of an organism shaped by its behavior and physiology, and this is in agreement with the Hutchinsonian niche concept. This may seem like semantics, but it has large implications for whether behavior and physiology, i.e., the organism, shape the realized niche or whether the realized niche shapes the organism, and thus this can influence the interpretation and discussion of the results. In general, the authors assume that behavior and physiology shape the realized niche throughout the manuscript. Personally, I think the concept of "niche" is challenging to use effectively because of the different definitions and dependence on scale (Kearney 2006; Soberón and Nakamura 2009) and that stronger arguments can be made by discussing behavior and physiology in the context of the relevant selective pressures. If the authors choose to keep "niche" as a focal topic in the manuscript, then the contradiction in the definition needs to be justified with citations if the authors believe both definitions are correct or resolved, and the manuscript should reflect the definition throughout.

Line 15: "a new concept:" should be removed. See references above.

Line 26: Maybe consider changing the phrasing. One could argue that adaptations resulting from evolution are not "predetermined".

Lines 43-45: Species do have fundamental niches shaped over deep-evolutionary time, but examples exist of species that can switch between phases of the diel cycle in response to extrinsic forces (e.g., Gaynor et al. 2018; Levy et al. 2019; Cox et al. 2021). This statement should be rephrased to reflect this variation.

Line 45-48: I agree! This is a strong aim of the paper.

Literature cited:

Cox DTC, Gardner AS, Gaston KJ. 2021. Diel niche variation in mammals associated with expanded trait space. *Nat Commun.* 12(1):1753. doi:10.1038/s41467-021-22023-4.

Gaynor KM, Hojnowski CE, Carter NH, Brashares JS. 2018. The influence of human disturbance on wildlife nocturnality. *Science.* 360(6394):1232–1235. doi:10.1126/science.aar7121.

Kearney M. 2006. Habitat, environment and niche: what are we modelling? *Oikos.* 115(1):186–191. doi:10.1111/j.2006.0030-1299.14908.x.

Kronfeld-Schor N, Dayan T. 2003. Partitioning of time as an ecological resource. *Annu Rev Ecol Evol Syst.* 34(1):153–181. doi:10.1146/annurev.ecolsys.34.011802.132435.

Kronfeld-Schor N, Tamar Dayan, Ralf Elvert, Abraham Haim, Nava Zisapel, Gerhard Heldmaier. 2001. On the use of the time axis for ecological separation: diel rhythms as an evolutionary constraint. *Am Nat.* 158(4):451–457. doi:10.1086/321991.

Levy O, Dayan T, Porter WP, Kronfeld-Schor N. 2019. Time and ecological resilience: can diurnal animals compensate for climate change by shifting to nocturnal activity? *Ecol Monogr.* 89(1):e01334. doi:10.1002/ecm.1334.

Soberón J, Nakamura M. 2009. Niches and distributional areas: Concepts, methods, and assumptions. *Proc Natl Acad Sci.* 106:19644–19650. doi:10.1073/pnas.0901637106.

Reviewer #2 (Remarks to the Author):

Here vertically migrating plankton are tracked using an acoustic system (ADCP) and the light levels at the depths of the vertically migrating layers are assessed using a light attenuation model and light levels above the surface. It is concluded that by vertically migrating, plankton dampen down changes to their ambient light level. This observation is self-evident but is still important to make.

This is a nice solid manuscript that I enjoyed reading. This topic has been extensively researched over the last 30 years, with much of the classic literature on animals tracking isolumens and depth selection dating back 20-30 years. The authors put a new spin on this topic, which may help generate renewed interest in this topic. I think this work will be solid addition to the literature. I think the manuscript needs better coverage of the older literature and some repetition needs to be deleted to make this piece more readable. With some revision, I think this will make a nice contribution.

1. Abstract. This is lacking in any factual results and a little vague. Include some figures. E.g. what is the variation in light level you predict for typical migrating bands versus those that would exist if they tracked an isolume ? i.e. how far do they stray from an isolume ? Do migrating bands always stay on the "dark side" of an isolume ? Provide some data.

2. Line 68. "This creates a paradox as the 'isolume-following' migration behavior should mask or even remove the overt diel light cycle (defined here as the realized light niche) which is considered the principle DVM trigger."

I am not sure there is really a paradox here. There is a big literature of older papers you need to tap into. Many have looked to see if plankton follow an isolume. It has generally been concluded that smaller plankton (e.g. classic migrators like the copepods *Metridia* and *Pleuromamma*) cannot swim fast enough to follow an isolume. So for them, of course, the light levels will change regardless of what they do. Generally, it is considered that it is the fast moving micro-nekton that can follow an isolume.

3. I think you need to discuss that the ADCP does not allow you to follow individuals and that individuals may be moving at very different speeds to the overall migrating band. Again there is older literature on this topic.

4. Linked to (2) I think you need to emphasise that future studies would benefit from net-validation of what is in the acoustic bands. This is a weakness of your work.

5. Introduction, line 62. I think you need to mention that daytime depth seems related to plankton size. Larger species tend to reside deeper in the daytime (i.e. darker) (e.g. "Buskey et al. (1989). Photosensitivity of the oceanic copepods *Pleuromamma* ... MEPS 55, 207-216") and spend less time near the surface each night (e.g. "Interspecific differences in diel vertical migration of marine copepods ... L&O 39, 1621-1629.).

6. Linked to Buskey reference above I think you need to mention that others have directly measured sub-surface light levels and acknowledge a weakness is that you have not directly measured light levels and this is an important goal for future work.

7. Can you collect some sub-surface light data to validate your light modelling work or somehow give some information on the likely errors this modelling introduces ? Or have others validated your approach ?

8. Line 56. "ascend to surface waters at night to feed on unicellular algae"
No. They are also feeding on microzooplankton (ciliates etc). Most of these migrating plankton are omnivores.

9. Line 27. Why only herbivores ?

10. Line 52. Some autotrophs move vertically, e.g. some dinoflagellates.

11. Line 57. This is normal DVM. Note that the reverse pattern (reverse DVM) is evident when invertebrate (non-visual) predators dominate.

12. Line 116. "... it is further possible that light intensity sets an upper limit rather than a specific target depth ..."

I think you need to quantify this aspect. Plot ambient light level for migrating bands versus time of day. Also plot depth versus time of day for migrating bands and also for each band the depth of the initial deep-daytime isolume versus time of day. So then the reader can see how far the migrating band lags behind the isolume.

13. Line 120. "The width of the scattering layer may however also be affected by the variation in species specific migration behaviors".

Note different individuals may migrate at variable speeds, so the leading edge of the layer is not the same individuals. Animals are typically moving in bursts.

14. Line 126. "The DVM behavior resulting from trade-off between food availability and predation risk thereby creates an RDLN to which the animals can adapt through phenotypic plasticity as well as evolutionarily.

I think you need to say that body condition modulates the relationship between food availability

and risk taking (E.g "Bruce Frost et al. (2001). Individual variability in diel vertical migration of a marine copepod *Limnology & Oceanography* 46, 2050-2054."). So I think you need to mention that sometime animals remain at depth, e.g. if they have sufficient body reserves (stored lipid for copepods) to avoid going near the surface. So animals tune the light levels they select through migration, or not, as part of the trade-off of foraging benefit versus predation risk.

15. Line 126. I would also mention that risk of predation modulate the behaviour of plankton, e.g. Bruce Frost et al. 1992. Variability of diel vertical migration in the marine planktonic copepod *Pseudocalanus newmani* in relation to its predators. *Can. J. Fish. Aquat. Sci.* 49: 1137-1 141.

16. Line 134. *Calanus* spp. do not exhibit strong DVM and certainly not between the surface and 200m.

17. Line 159-191. Wordy repetition. Delete.

18. Line 199. "The biomass of the zooplankton community is strongly dominated by euphausiids (krill) of the genus *Thysanoessa* with additional significant contribution by copepods of the genus *Calanus* (*C. finmarchicus* and *C. glacialis*). Both groups are well known to perform DVM in the fjord."

Give more detail. *Calanus* does not exhibit strong DVM.

19. Lines 212. "The phytoplankton concentration at the time was negligible and was not considered for underwater light calculations."

Clarify what you mean by negligible. So why come to the surface to feed at all ? See earlier comments about being clear about the limitations of your modelling.

20. Light levels are modelled from measurements in the air. This is a limitation. Can you complete some validation trials of your modelling ?

21. Figure 1. Also include time on the x-axis, so the reader can see if the dates "20 Mar, 21 Mar etc) are indicated at the start of each day or at midday.

22. Also plot the isolume of the deep daytime depth.

In summary, I think this is a solid manuscript. With careful revision, I think this will make a nice contribution and congratulate the authors on completing such a nice piece of work.

Reviewer #3 (Remarks to the Author):

In the manuscript "Animal behavior is central in shaping the realized light niche" submitted to *Communications Biology*, Häfker et al. demonstrate how an animal's realized daylight niche is mutually defined by environmental dial fluctuations and the animal's behavioral adaptation to such changes. The study is interesting as it aims to describe environmental shifts from an animal's perspective. In the case of zooplankton, the study shows that the daily fluctuations in light intensity are much milder from the animal's perspective and that wavelength spectrum instead of light intensity may be important for regulating the circadian clock.

The manuscript presents a study design that is well suited to answer the research question, and I have no major concerns regarding the scientific backbone of this exciting and important study. However, when reading through the manuscript, I find that it is sometimes confusing to tell apart exactly what is concluded from the study, what is the authors' assumptions, and what is shown in previous studies. I suggest some minor edits that, if implemented, will help the reader's orientation in the study. Numbers refer to line numbers.

15: Add comma: "habitat,"

18: When you write "Highly dampened light field ", do you mean rather "reduced light field fluctuations"?

24: Throughout the introduction, I find it difficult to navigate between the background, your assumptions, and your hypothesis. Consider being more explicit when you describe what can be expected from this study. Expressions like "In this study we..." or "Here we..." are often helpful. I will give some suggestions below:

36: I do not think it is clear what you mean with "We propose...". Is this an assumption from your side, or is this a hypothesis that you aim to test with the study? Is the assumption yours, or is it described by someone else (it is followed by a reference)? Please be explicit in your position.

46: Here, you could emphasize that this is indeed the purpose of this study. "In this study, we aim...." or similar.

73: At this point, when I read the introduction, it is still very unclear to me exactly what your study is about. You write "Addressing these questions..." but would it be possible to be more explicit about -how- you are going to address it?

76: Here, you transition from introduction to results. I strongly suggest spelling that out in a heading to make it easier to navigate in the text.

77-78 In this manuscript format, methods are placed at the end of the manuscript. This means that the reader should be able to follow the results and discussion without having to read the methods part at the end. I think you need to help the reader and briefly explain what your approach is (without mentioning any technical details). One way of making this smooth could be to formulate a clear hypothesis in the introduction.

87: I get stuck in the flow of this sentence. Could you instead write: "... it is evident that changes in RDLN at the COM..."

112: I do not understand this sentence: "cycling of the RDLN is realized in DVM". It feels problematic that "realized" occurs twice here.

133: Here, you mention Kungsfjorden for the first time. If you want to name it, please introduce it when you introduce your study site, at the beginning of the results part.

210: This sentence starts in present tense. Is there a reason for that? I also find the claim "Data analysis focuses on" slightly cryptic. I think it would be clearer if you stated: "We focused our study on the week..."

212: I think it is philosophically questionable that the phytoplankton concentration is negligible. I think you should state this as your assumption, for instance, "As phytoplankton concentrations were low, we decided to exclude..." or similar.

219: Add comma: "Above the surface, ..."

Except for these minor remarks, I find the manuscript well-written and interesting to read. I congratulate the Authors on the good work done so far and wish you good luck with publishing this paper that will benefit the biological research community.

Dear reviewers,

We are happy that you appreciate our manuscript and are very thankful for your valuable comments. We are also very pleased that the reviewers thought the manuscript provides an interesting and novel general contribution to the literature. We agree with the vast majority of the comments and have articulated below the specific changes made to address them, and justified the instances where we differed from reviewer opinion. Please find below our responses to the individual comments (indented and in red text).

Reviewer #1 (Remarks to the Author):

Summary:

The authors postulate that animals' realized vs. fundamental niche has a strong light component and that the realized light niche is shaped by animal behavior. The authors suggest that to understand the role of light in the realized niche, that the perspective of the organism should be taken. They use a marine zooplankton community to demonstrate that due to diel vertical migration (DVM) that factors experienced as part of the light environment by the zooplankton is rather similar or opposite than expected across the diel cycle depending on depth. The authors conclude that the DVM behavior of the zooplankton affects their realized light niche.

Impression:

I really enjoyed reading this manuscript because the writing was clear and concise. I believe it is a novel contribution to the literature, specifically the concept of taking the viewpoint of the animal's perception of light across the diel cycle. This angle is rarely taken in empirical investigations, especially in the field, and can be useful by many disciplines, including evolutionary biology, ecology, physiology, and chronobiology. This paper specifically highlights the need to incorporate cues important to the organism being investigated to better understand physiological mechanism and ecological process. Altogether, this manuscript would be a great contribution to the literature if relatively few issues were resolved.

Data & methodology, including statistics:

There is no data availability statement, and it is unclear whether the calculation of COM is a result of programming using code or a specific program. The supplied data appear to be good quality and seems efficient to repeat the analyses.

A data availability statement was added at the end of the methods section (line 327-329). Calculation of the COM is based on the paper by Urmy et al (2012) and does not require any particular software. We expanded the respective part in the methods to make the origin of the calculations more obvious (line 284).

I am unable to evaluate fully the marine zooplankton community and marine irradiance measurements because these methods are outside my expertise. The majority of the methods presented are following previously published methods.

Lomb-Scargle periodogram analysis is an appropriate method for identifying rhythmicity in ecological data.

Comments:

Although, I believe that the realized diel light niche (RDLN) is a useful term in the context of this manuscript, the term draws a strong analogy to previous discussion on the temporal niche, in which animal behavior and species interactions directly influence the light experienced by an organism. The following articles are some seminal discussions on this topic that should be incorporated into the manuscript: Kronfeld-Schor et al. 2001; Kronfeld-Schor and Dayan 2003.

The references were included to highlight the effects of behavior in the context of fundamental/realized niches as well as circadian clocks and their functioning/evolution (line 55-63, 244-248).

There seems to be some logical inconsistencies in whether the realized niche is a property of an organism resulting from their adaptations (lines 43-45) or is a group of selective forces acting on an organism resulting in adaptations. The definition of “realized niche” in the manuscript states that organisms need to be (i.e., must be) adapted to the realized niche (lines 41-43). Kearney (2006) defines the realized niche as a property of an organism shaped by its behavior and physiology, and this is in agreement with the Hutchinsonian niche concept. This may seem like semantics, but it has large implications for whether behavior and physiology, i.e., the organism, shape the realized niche or whether the realized niche shapes the organism, and thus this can influence the interpretation and discussion of the results. In general, the authors assume that behavior and physiology shape the realized niche throughout the manuscript. Personally, I think the concept of “niche” is challenging to use effectively because of the different definitions and dependence on scale (Kearney 2006; Soberón and Nakamura 2009) and that stronger arguments can be made by discussing behavior and physiology in the context of the relevant selective pressures. If the authors choose to keep “niche” as a focal topic in the manuscript, then the contradiction in the definition needs to be justified with citations if the authors believe both definitions are correct or resolved, and the manuscript should reflect the definition throughout.

We rephrased the section for a better distinction of fundamental and realized niche. We agree with the definition by Kearney that the realized niche is shaped by physiology and behavior in response to the fundamental niche and interaction with other animals. We also included Kearney 2006 as a reference (line 42-53). We do, however, also point out that adaptation should not be limited to the fundamental niche. For example, diel cycles of light sensitivity under the control of a circadian clock mostly makes sense to prevent unwanted sensitivity adaptations that otherwise would arise from short-term changes in light conditions due to an animal e.g. entering/leaving burrows or strongly shaded areas (e.g. line 181-187).

Line 15: “a new concept:” should be removed. See references above.

Changed as suggested (line 16-17).

Line 26: Maybe consider changing the phrasing. One could argue that adaptations resulting from evolution are not “predetermined”.

The part was rephrased to make clear that it is the 'optimal time of the day' that depends on the environment (line 29).

Lines 43-45: Species do have fundamental niches shaped over deep-evolutionary time, but examples exist of species that can switch between phases of the diel cycle in response to extrinsic forces (e.g., Gaynor et al. 2018; Levy et al. 2019; Cox et al. 2021). This statement should be rephrased to reflect this variation.

This aspect was included as suggested (line 55-61).

Line 45-48: I agree! This is a strong aim of the paper.

Literature cited:

Cox DTC, Gardner AS, Gaston KJ. 2021. Diel niche variation in mammals associated with expanded trait space. *Nat Commun.* 12(1):1753. doi:10.1038/s41467-021-22023-4.

Gaynor KM, Hojnowski CE, Carter NH, Brashares JS. 2018. The influence of human disturbance on wildlife nocturnality. *Science.* 360(6394):1232–1235. doi:10.1126/science.aar7121.

Kearney M. 2006. Habitat, environment and niche: what are we modelling? *Oikos.* 115(1):186–191. doi:10.1111/j.2006.0030-1299.14908.x.

Kronfeld-Schor N, Dayan T. 2003. Partitioning of time as an ecological resource. *Annu Rev Ecol Evol Syst.* 34(1):153–181. doi:10.1146/annurev.ecolsys.34.011802.132435.

Kronfeld-Schor N, Tamar Dayan, Ralf Elvert, Abraham Haim, Nava Zisapel, Gerhard Heldmaier. 2001. On the use of the time axis for ecological separation: diel rhythms as an evolutionary constraint. *Am Nat.* 158(4):451–457. doi:10.1086/321991.

Levy O, Dayan T, Porter WP, Kronfeld-Schor N. 2019. Time and ecological resilience: can diurnal animals compensate for climate change by shifting to nocturnal activity? *Ecol Monogr.* 89(1):e01334. doi:10.1002/ecm.1334.

Soberón J, Nakamura M. 2009. Niches and distributional areas: Concepts, methods, and assumptions. *Proc Natl Acad Sci.* 106:19644–19650. doi:10.1073/pnas.0901637106.

Reviewer #2 (Remarks to the Author):

Here vertically migrating plankton are tracked using an acoustic system (ADCP) and the light levels at the depths of the vertically migrating layers are assessed using a light attenuation model and light levels above the surface. It is concluded that by vertically migrating, plankton dampen down changes to their ambient light level. This observation is self-evident but is still important to make.

This is a nice solid manuscript that I enjoyed reading. This topic has been extensively researched over the last 30 years, with much of the classic literature on animals tracking isolumes and depth selection dating back 20-30 years. The authors put a new spin on this topic, which may help generate renewed

interest in this topic. I think this work will be solid addition to the literature. I think the manuscript needs better coverage of the older literature and some repetition needs to be deleted to make this piece more readable. With some revision, I think this will make a nice contribution.

We agree with the reviewer that DVM encompasses various different aspects that we do not cover in detail (e.g. the effect of body size and condition, reverse DVM/midnight sinking, migration speeds of different species including phytoplankton). With this manuscript we aim to highlight the effects of behavior on the experienced environment not only for marine pelagic communities but in general, which is why we decided on the 'Perspective' format. Thus, we are concerned that exploring too many different aspects of DVM in great detail would steer focus away from the overall relevance for broader scientific investigations, which we discuss at the end of the manuscript. In the revision, we now provide more detail on individual aspects of DVM, but still tried to make the story accessible to a broad readership that may not be familiar with zooplankton communities.

1. Abstract. This is lacking in any factual results and a little vague. Include some figures. E.g. what is the variation in light level you predict for typical migrating bands versus those that would exist if they tracked an isolume ? i.e. how far do they stray from an isolume ? Do migrating bands always stay on the "dark side" of an isolume ? Provide some data.

We now provide more specific details in the abstract. As discussed above, we avoid an exclusive focus on the isolume as the implications of our work are not limited to plankton communities.

2. Line 68. "This creates a paradox as the 'isolume-following' migration behavior should mask or even remove the overt diel light cycle (defined here as the realized light niche) which is considered the principle DVM trigger."

I am not sure there is really a paradox here. There is a big literature of older papers you need to tap into. Many have looked to see if plankton follow an isolume. It has generally been concluded that smaller plankton (e.g. classic migrators like the copepods *Metridia* and *Pleuromamma*) cannot swim fast enough to follow an isolume. So for them, of course, the light levels will change regardless of what they do. Generally, it is considered that it is the fast moving micro-nekton that can follow an isolume.

The phrasing was toned down to make it clearer that perfect isolume-following is not a given (line 92-95). However, the fact persists that isolume-following should minimize the diel light cycle (even if it is not completely eliminated), which seems somewhat contradictory to light being the central cue for DVM and also for the synchronization of endogenous clocks and other (e.g. metabolic) rhythms. As discussed above, we believe that going into great detail about the migrations of individual species or a thorough discussion of our investigation in the context of the isolume would go beyond the scope of our work.

3. I think you need to discuss that the ADCP does not allow you to follow individuals and that individuals may be moving at very different speeds to the overall migrating band. Again there is older literature on this topic.

We expanded the discussion of the diversity within the zooplankton community and now mention species- and individual-specific migration patterns, reverse DVM and midnight

sinking, the influence of body size, as well as how these factors relate to the COM and RDLNs (line 163-175).

4. Linked to (2) I think you need to emphasise that future studies would benefit from net-validation of what is in the acoustic bands. This is a weakness of your work.

The need for and benefits of net-samplings were added to the methods section (line 280-282).

5. Introduction, line 62. I think you need to mention that daytime depth seems related to plankton size. Larger species tend to reside deeper in the daytime (i.e. darker) (e.g. “Buskey et al. (1989). Photosensitivity of the oceanic copepods *Pleuromamma* MEPS 55, 207-216”) and spend less time near the surface each night (e.g. “Interspecific differences in diel vertical migration of marine copepods ... L&O 39, 1621-1629.).

Please see response to comment 3.

6. Linked to Buskey reference above I think you need to mention that others have directly measured sub-surface light levels and acknowledge a weakness is that you have not directly measured light levels and this is an important goal for future work.

A sentence mentioning the benefits of using sub-surface measurements as the basis for modeling was added at the end of the respective methods section (line 324-325).

7. Can you collect some sub-surface light data to validate your light modelling work or somehow give some information on the likely errors this modelling introduces ? Or have others validated your approach ?

The method for modeling underwater light conditions was validated using Hydrolight simulations as well as in situ measurements by the original authors (Pan & Zimmermann 2010). and now explicitly state this in the method part. Furthermore we added a sentence stating that currently available sensors do lack the sensitivity to generate comparable data (lines 314-317).

8. Line 56. “ascend to surface waters at night to feed on unicellular algae” No. They are also feeding on microzooplankton (ciliates etc). Most of these migrating plankton are omnivores.

We removed ‘on unicellular algae’ to include all possible food sources in surface waters (line 81)

9. Line 27. Why only herbivores ?

The respective reference (Benhamou 2014) strongly focuses on herbivores, but as the concept applies to all mobile animals, we agree and changed the text accordingly (line 30).

10. Line 52. Some autotrophs move vertically, e.g. some dinoflagellates.

We decided not to include this aspect as phytoplankton migrations are limited to comparably small depth changes (e.g. Sournia 1975. Circadian Periodicities in Natural

Populations of Marine Phytoplankton. doi: 10.1016/S0065-2881(08)60460-5). To keep the story concise (see also responses above) no changes were made.

11. Line 57. This is normal DVM. Note that the reverse pattern (reverse DVM) is evident when invertebrate (non-visual) predators dominate.

Please see response to comment 3.

12. Line 116. "... it is further possible that light intensity sets an upper limit rather than a specific target depth ..."

I think you need to quantify this aspect. Plot ambient light level for migrating bands versus time of day. Also plot depth versus time of day for migrating bands and also for each band the depth of the initial deep-daytime isolume versus time of day. So then the reader can see how far the migrating band lags behind the isolume.

We do plot depth (COM) and light versus time in Fig.1B,C. It is not possible to resolve the backscatter data into individual bands that could be compared to an isolume (Fig.1A). A separate paper (Hobbs et al. 2021) thoroughly investigates the relationship of scattering layers and the isolume and we cite the paper in this section. Hence, we elect to not pursue this aspect greater detail. Please also see our response regarding the overall focus of the manuscript.

13. Line 120. "The width of the scattering layer may however also be affected by the variation in species specific migration behaviors".

Note different individuals may migrate at variable speeds, so the leading edge of the layer is not the same individuals. Animals are typically moving in bursts.

The part was expanded to emphasize the difference between the community- and individual-level RDLN (line 163-175).

14. Line 126. "The DVM behavior resulting from trade-off between food availability and predation risk thereby creates an RDLN to which the animals can adapt through phenotypic plasticity as well as evolutionarily.

I think you need to say that body condition modulates the relationship between food availability and risk taking (E.g "Bruce Frost et al. (2001). Individual variability in diel vertical migration of a marine copepod *Limnology & Oceanography* 46, 2050-2054."). So I think you need to mention that sometime animals remain at depth, e.g. if they have sufficient body reserves (stored lipid for copepods) to avoid going near the surface. So animals tune the light levels they select through migration, or not, as part of the trade-off of foraging benefit versus predation risk.

We included additional references (incl. Hay et al. 2001 and Frost et al. 1992) to provide readers with detailed investigations of how phenotype and evolution interact with DVM and RDLNs (line 188-190). We believe that discussing these points in detail here would go beyond the scope of the manuscript. Frost et al. 1992 was also included in line 167.

15. Line 126. I would also mention that risk of predation modulate the behaviour of plankton, e.g. Bruce Frost et al. 1992. Variability of diel vertical migration in the marine planktonic copepod *Pseudocalanus newmani* in relation to its predators. *Can. J. Fish. Aquat. Sci.* 49: 1137-1141.

Please see response to comment 14.

16. Line 134. *Calanus* spp. do not exhibit strong DVM and certainly not between the surface and 200m.

At this point we discuss light sensitivity and not DVM. Nevertheless, *Calanus* spp. does show DVM that can span over more than 100 m, also in Svalbard fjords (e.g. Häfker et al. 2018. doi: 10.1002/lno.11011, Daase et al. 2008. doi: 10.1080/17451000801907948). No changes were made.

17. Line 159-191. Wordy repetition. Delete. .

We feel the respective section puts our specific findings in the context of laboratory experiments and study design in general. The second half of the indicated section is now clearly identified as conclusions for which some repetition is unavoidable.

18. Line 199. “The biomass of the zooplankton community is strongly dominated by euphausiids (krill) of the genus *Thysanoessa* with additional significant contribution by copepods of the genus *Calanus* (*C. finmarchicus* and *C. glacialis*). Both groups are well known to perform DVM in the fjord.”
Give more detail. *Calanus* does not exhibit strong DVM.

Please see response to comment 16.

19. Lines 212. “The phytoplankton concentration at the time was negligible and was not considered for underwater light calculations.”

Clarify what you mean by negligible. So why come to the surface to feed at all ? See earlier comments about being clear about the limitations of your modelling.

The sentence was rephrased (line 293-295, also in response to comment by reviewer 3). DVM in this low-chlorophyll environment may represent an adaptation to the variability in spring bloom timing to avoid ‘missing out’ on the start of the bloom and a corresponding sentence was added (line 110-112).

20. Light levels are modelled from measurements in the air. This is a limitation. Can you complete some validation trials of your modelling ?

The respective part was expanded to provide more information on the validation of the model (line 314-317, 320, 324-325).

21. Figure 1. Also include time on the x-axis, so the reader can see if the dates “20 Mar, 21 Mar etc) are indicated at the start of each day or at midday.

The corresponding information was added to the legend by stating that date ticks indicate midnight of the respective days. The same information was also added to the legends of Fig.2 and Fig.S3. Dates were also added to panels A and B of Fig.1.

22. Also plot the isolume of the deep daytime depth.

We cannot define a specific threshold light sensitivity and a corresponding isolume for the zooplankton community. Sensitivity measurement for *Thysanoessa* and *Calanus* exist, but

the thresholds differ between species. They were further determined during the polar night season and are not necessarily representative for the spring equinox. Thus we believe it is more correct to plot the complete underwater light field as done in Fig.1B.

In summary, I think this is a solid manuscript. With careful revision, I think this will make a nice contribution and congratulate the authors on completing such a nice piece of work.

Reviewer #3 (Remarks to the Author):

In the manuscript “Animal behavior is central in shaping the realized light niche” submitted to Communications Biology, Häfker et al. demonstrate how an animal’s realized daylight niche is mutually defined by environmental dial fluctuations and the animal’s behavioral adaptation to such changes. The study is interesting as it aims to describe environmental shifts from an animal’s perspective. In the case of zooplankton, the study shows that the daily fluctuations in light intensity are much milder from the animal’s perspective and that wavelength spectrum instead of light intensity may be important for regulating the circadian clock.

The manuscript presents a study design that is well suited to answer the research question, and I have no major concerns regarding the scientific backbone of this exciting and important study. However, when reading through the manuscript, I find that it is sometimes confusing to tell apart exactly what is concluded from the study, what is the authors' assumptions, and what is shown in previous studies. I suggest some minor edits that, if implemented, will help the reader’s orientation in the study. Numbers refer to line numbers.

We reworked the manuscripts focusing on a clearer distinction of the different sections and the use of indicators like ‘Here we...’ to make it more structured and easier to read. We also added more line-breaks as well as section headings more better orientation.

15: Add comma: “habitat,”

Changed as suggested (line 16).

18: When you write “Highly dampened light field “, do you mean rather “reduced light field fluctuations”?

The sentence was rephrased to make it clearer (line 19-21).

24: Throughout the introduction, I find it difficult to navigate between the background, your assumptions, and your hypothesis. Consider being more explicit when you describe what can be expected from this study. Expressions like “In this study we...” or “Here we...” are often helpful. I will give some suggestions below:

While we do not explicitly state any expectations regarding the results, we reworked the end of the introduction to more clearly state our objectives (line 68-72). Please also see response above.

36: I do not think it is clear what you mean with “We propose...”. Is this an assumption from your side, or is this a hypothesis that you aim to test with the study? Is the assumption yours, or is it described by someone else (it is followed by a reference)? Please be explicit in your position.

The sentence was rephrased making clear that this is not a hypothesis, but that we explore the feedback between behavior and environment in the context of fundamental vs. realized niche theory (line 39-41).

46: Here, you could emphasize that this is indeed the purpose of this study. “In this study, we aim...” or similar.

We reworked the end of the introduction to clearly state our objectives and the topics we discuss (line 68-72). Please also see response above.

73: At this point, when I read the introduction, it is still very unclear to me exactly what your study is about. You write “Addressing these questions...” but would it be possible to be more explicit about - how- you are going to address it?

We now clearly state our focus at the end of the introduction (see previous comment) and also reworked the indicated sentence to provide reasoning on why we decided to investigate RDLNs in a marine zooplankton community (line 99-104).

76: Here, you transition from introduction to results. I strongly suggest spelling that out in a heading to make it easier to navigate in the text.

We added headings to the sections for better structuring. We also reworked the beginning of the section to more clearly indicate that now results are presented. We think that using ‘Results’ as a headings would not fit the style of a Perspective-paper.

77-78 In this manuscript format, methods are placed at the end of the manuscript. This means that the reader should be able to follow the results and discussion without having to read the methods part at the end. I think you need to help the reader and briefly explain what your approach is (without mentioning any technical details). One way of making this smooth could be to formulate a clear hypothesis in the introduction.

We added a few more details to the method description in the main text (line 107-118, 122-125) to make it more readily understandable. We further reworked to the beginning of the section to provide a better overview of our study approach.

87: I get stuck in the flow of this sentence. Could you instead write: “... it is evident that changes in RDLN at the COM...”

Changed as suggested (line 126).

112: I do not understand this sentence: “cycling of the RDLN is realized in DVM”. It feels problematic that “realized” occurs twice here.

The sentence was reworked (line 153).

133: Here, you mention Kungsfjorden for the first time. If you want to name it, please introduce it when you introduce your study site, at the beginning of the results part.

We now also mention Kongsfjorden at the beginning of the third section where we described the study site during the investigation period (line 108).

210: This sentence starts in present tense. Is there a reason for that? I also find the claim "Data analysis focuses on" slightly cryptic. I think it would be clearer if you stated: "We focused our study on the week...".

Changed as suggested (line 291).

212: I think it is philosophically questionable that the phytoplankton concentration is negligible. I think you should state this as your assumption, for instance, "As phytoplankton concentrations were low, we decided to exclude..." or similar.

We understand the concern and rephrased the sentence (line 293-295).

219: Add comma: "Above the surface, ..."

'Above surface spectral irradiance' was meant to be read as one term. We rephrased it to make it more understandable (line 301).

Except for these minor remarks, I find the manuscript well-written and interesting to read. I congratulate the Authors on the good work done so far and wish you good luck with publishing this paper that will benefit the biological research community.